# Ultrasmall and tunable TeraHertz surface plasmon cavities at the ultimate plasmonic limit

Ian Aupiais[1] ✉, Romain Grasset [1], Tingwen Guo[1], Dmitri Daineka[2], Javier Briatico [3], Sarah Houver[4], Luca Perfetti [1], Jean-Paul Hugonin[5], Jean-Jacques Greffet [5] & Yannis Laplace[1] ✉

The ability to confine THz photons inside deep-subwavelength cavities promises a transformative impact for THz light engineering with metamaterials and for realizing ultrastrong light-matter coupling at the single emitter level. To that end, the most successful approach taken so far has relied on cavity architectures based on metals, for their ability to constrain the spread of electromagnetic fields and tailor geometrically their resonant behavior. Here, we experimentally demonstrate a comparatively high level of confinement by exploiting a plasmonic mechanism based on localized THz surface plasmon modes in bulk semiconductors. We achieve plasmonic confinement at around 1 THz into record breaking small footprint THz cavities exhibiting mode volumes as low as $V_{cav}/\lambda_0^3 \sim 10^{-7} - 10^{-8}$, excellent coupling efficiencies and a large frequency tunability with temperature. Notably, we find that plasmonic-based THz cavities can operate until the emergence of electromagnetic nonlocality and Landau damping, which together constitute a fundamental limit to plasmonic confinement. This work discloses nonlocal plasmonic phenomena at unprecedentedly low frequencies and large spatial scales and opens the door to novel types of ultrastrong light-matter interaction experiments thanks to the plasmonic tunability.

Due to their functional relevance in many fields, tremendous efforts have been devoted in the past years to the development of optical cavities in the TeraHertz (THz) frequency range. As for their optical counterparts, THz cavities operating at subwavelength scales offer many appealing prospects ranging from the control of THz light fields with metamaterials[1–7] to the potential realization of new phases of matter by engineering ultrastrong light-matter coupling at low energies in condensed matter systems[8–12]. To date, the most successful strategy pursued to confine low frequency THz fields into deep-subwavelength volumes has been based on the design of metallic

resonators. *Photonic* cavities for instance, which rely on the quasi-free space propagation of electromagnetic (EM) fields and are typically diffraction limited in 3D[13,14], can be made highly subwavelength by restricting EM propagation to a single spatial dimension in metal/insulator/metal THz microcavities[15,16]. Similarly, *electronic* cavities, such as LC resonators, make use of the capacitance and geometric inductance of metals to tailor the resonant behavior of EM-fields with design principles from electronics[2,17]. They operate at subwavelength scales in all three dimensions and have demonstrated ultrastrong light-matter coupling at THz frequencies[18–20]. More recently, 3D lumped

[1]LSI, CEA/DRF/IRAMIS, CNRS, Ecole Polytechnique, Institut Polytechnique de Paris, Palaiseau, France. [2]LPICM, CNRS, Ecole Polytechnique, Institut Polytechnique de Paris, Palaiseau, France. [3]Unité Mixte de Physique, CNRS, Thales, Université Paris Saclay, 91767 Palaiseau, France. [4]Université Paris Cité, CNRS, Matériaux et Phénomènes Quantiques, F-75013 Paris, France. [5]Université Paris-Saclay, Institut d'Optique Graduate School, CNRS, Laboratoire Charles Fabry, 91127 Palaiseau, France. ✉e-mail: ian.aupiais@unige.ch; yannis.laplace@polytechnique.edu

resonator architectures have emerged that have set new records regarding the smallest achievable dimensions for THz resonators[21,22]. At last, *plasmonic* cavities[23–25] are those which exploit surface plasmons and the kinetic inductance of free or bound charges for light confinement[26]. In contrast with the previous two approaches, metals are ineffective in providing plasmonic confinement at low THz frequencies whereas semiconductors, thanks to their low carrier densities, support surface plasmons in the THz frequency range. In this respect, narrow-gap semiconductors such as InSb, InAs and HgCdTe have long been recognized of special importance in the context of THz plasmonics thanks to the low effective mass of their carriers, resulting in high carrier mobilities and large plasmonic tunability with parameters such as temperature[27,28], doping[29], dynamic[30,31] and static[27,29,32] EM-fields. Surface plasmonic related phenomena in these materials have been studied experimentally and theoretically in various contexts for sensing applications[33–35], as passive and active plasmonic platforms[36–41], for thermal emission[42] and THz detection[43].

Despite this rapid progress, little is known experimentally regarding the ultimate degree at which a low frequency THz light field can efficiently be confined in cavities operating with a surface plasmonic mechanism and whether such cavities can be competitive or even advantageous with respect to their metallic-based counterparts. Fundamentally, the ultimate limit to plasmonic confinement manifests via frequency-shifts and broadening of the plasmonic resonances due to the onset of EM-nonlocality and collisionless Landau damping[44,45] (see Fig. 1a). In practice however, resonance broadening originating from pervasive material loss may render these effects unobservable and this limit difficult to reach. To date, it was observed experimentally in metals at optical frequencies[44,46] and at frequencies above 30 THz in graphene[47,48]. At lower frequencies, the ability to reach such a fundamental limit remains an open question.

In order to address the ultimate limits of EM-confinement in plasmonic-based THz cavities, we manufactured THz cavities using the narrow-gap bulk semiconductor InSb as the THz plasmonic material. In Fig. 1b, we present the temperature evolution of the THz plasma edge, from which we determine in Fig. 1c the plasma frequency $\nu_p$, tunable from 0.5 to 2.3 THz, and the carrier scattering rate $\Gamma$ for temperatures between 200 K and 310 K (see methods and supplementary information). As shown in Fig. 1c, we observe that in the temperature and frequency ranges of interest, $\Gamma < \nu \lesssim \nu_p$ which characterizes the plasmonic regime of light-matter interaction[49].

Below, we experimentally demonstrate deep-subwavelength and widely temperature-tunable THz cavities based on a surface plasmonic mechanism. For the smallest cavities, we find that the plasmonic resonances exhibit strong signatures of EM-nonlocality and Landau damping, establishing that the fundamental limit to plasmonic confinement can be reached experimentally at frequencies as low as 1 THz. In this regime, THz plasmonic cavities are found to be competing alternatives to metallic-based cavities and may provide further advantages thanks to the plasmonic tunability.

## Results

### THz plasmonic cavity architecture and experimental results at room temperature

The design of our cavities, depicted in Fig. 2a, is inspired by the metal/insulator/metal ("**m/i/m**") cavities widely investigated from the THz to the visible frequency ranges[50–52]. The key point of our work is to use the THz plasma of intrinsic InSb as a substitute for one of the metallic part in the cavity architecture, resulting in metal/insulator/plasma ("**m/i/p**") cavities. Samples consist in the bulk THz plasmonic material, a dielectric insulator of highly subwavelength thickness ($Si_3N_4$, thickness $d$) and a metallic stripe grating deposited on top (stripe width $s$ and spacing $a$). This corresponds to a periodic lattice of 'stripe' **m/i/p** cavities which, for all practical purposes, can be considered infinite in the $x$-$z$

plane (see methods). In the region below a given metallic stripe, the waveguide-like propagation of EM-waves (along the $x$-direction in Fig. 2a) originates from the coupling across the insulator of the surface plasmons at the two m/i and i/p interfaces. The finite length $s$ of the **m/i/p** waveguide leads to the cavity resonance. In practice, cavity frequency tuning is achieved by varying the stripe width $s$ while parameters $d$ and $a$, which control the radiative decay rate, are chosen to favor critical coupling of the cavity to external radiation[16].

We present in Fig. 2b the experimental reflectivity spectra obtained at 290 K for three cavity samples ($s = 21\,\mu m$, $31\,\mu m$, $41\,\mu m$) together with our parameter free EM-simulation computed from the experimentally determined optical constants and geometrical parameters of the constitutive elements (see methods and supplementary information). The reflectivity of bulk InSb at the same temperature is also shown for comparison. We observe a first resonance at low frequency evidenced by a pronounced absorption in the reflectivity spectra indicated by the circles in Fig. 2b. It is the fundamental resonance of the **m/i/p** cavity acting as a $\frac{\lambda}{2}$ resonator and the corresponding mode has a wavevector $k_x = \frac{\pi}{s}$. It is spatially localized under the metallic stripe and has a substantial fraction of its volume residing in the plasma, as seen from its profile displayed in Fig. 2d. The total power absorption of the resonance reaches values beyond 90% for the three designed cavities: hence, at this temperature and for the structural parameters we used, the cavities are essentially critically coupled to free-space THz fields[53]. A second, much less pronounced, resonance (triangles in Fig. 2b) is observed and corresponds to the third mode of the **m/i/p** cavity acting as a $\frac{3\lambda}{2}$ resonator. As expected, its mode profile exhibits three extrema of the magnetic field (Fig. 2d) which corresponds to a wavevector $k_x = \frac{3\pi}{s}$. Let's note that free space excitation of the second cavity mode is forbidden at normal incidence by symmetry[50]. A third sizeable resonance (squares in Fig. 2b) is seen to lie very close, albeit below, the transparency edge of the material as can be noticed from the comparison with the bulk plasma edge (Fig. 2b, black solid line). As shown in Fig. 2d, this mode is spatially localized between two neighboring cavities (spacing $a$) in the **vacuum/i/p** region and we refer to it as the "inter-cavity" mode. Having a finite penetration depth $\delta$ inside the neighboring cavities, the wavevector of this mode can be expressed as $k_x = \frac{\pi}{(a + 2\delta)}$ (see supplementary information).

In Fig. 2c, we report the theoretical dispersion relations for two infinite waveguides of the type **m/i/p** and **vacuum/i/p**. They are compared with the measured resonance frequencies $\nu_c$ plotted as a function of the mode wavevector $k_x$ for the set of geometrical parameters $s$ and $a$ studied: $k_x = \frac{\pi}{s}, \frac{3\pi}{s}, \frac{\pi}{(a + 2\delta)}$ for the fundamental, third and inter-cavity modes respectively. The overall picture that emerges from the agreement between the two is that both the cavity and inter-cavity regions act as Fabry-Perot resonators for THz surface plasmon modes. The scaling of their resonance frequencies with geometrical parameters ensues directly from the dispersion relations of the propagating modes of the corresponding **m/i/p** or **vacuum/i/p** plasmonic waveguides. Interestingly, we note that the inter-cavity mode exhibits a negative dispersion, an aspect which can be traced back to the symmetry of its mode profile. A comprehensive analysis of the modes' characteristics is given in the supplementary information.

### Temperature tuning and scaling properties of the THz plasmonic cavities

We now turn to the evolution of the plasmonic behaviour of the cavities as a function of the bulk THz plasma frequency $\nu_p$. To that end, we use temperature as the tuning parameter allowing $\nu_p$ to be varied continuously between 0.5 THz and 2.3 THz. In Fig. 3, we present the experimental and simulated reflectivity spectra of the three samples (Fig. 3a-c) which show a remarkably good agreement together. At large

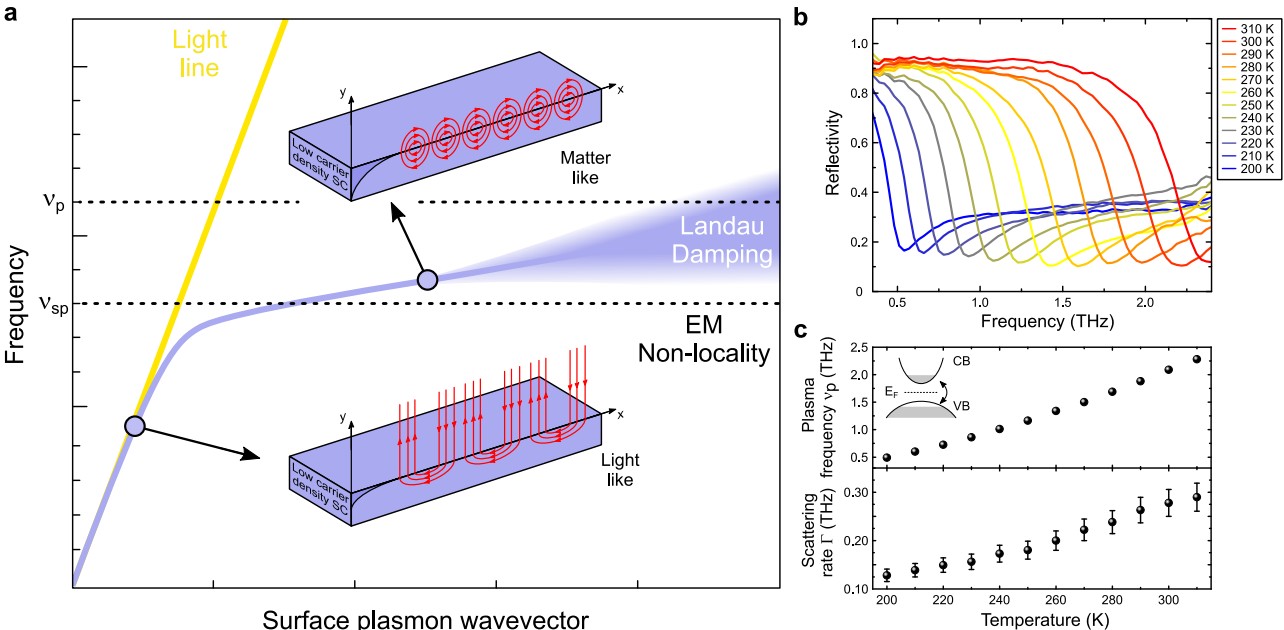

**Fig. 1 | Ultimate limits of light confinement with surface plasmons and THz plasmonic properties of bulk narrow-gap semiconductor InSb. a** Light confinement mechanism and its ultimate limit with surface plasmons. At frequencies $\nu_{sp}$ close to the plasma frequency $\nu_p$ of the semiconductor, the dispersion relation of the surface plasmon (solid purple line) strongly departs from that of the light (solid yellow line) in the transparent medium adjacent to the interface, allowing for confinement of the EM-fields both longitudinally (along the direction of propagation) and transversally to the interface. The fundamental limit to plasmonic confinement is signaled through EM-nonlocality via the departure of the dispersion relation from its asymptotic value $\nu_{sp}$, as well as the progressive vanishing of the surface plasmon resonance due to collisionless Landau damping. In practice, these effects are usually obscured by the intrinsic material losses such as carrier scattering. **b** THz reflectivity spectra of bulk InSb as a function of temperature between 200 K and 310 K exhibiting a sharp THz plasma edge that is tunable with the temperature. Below the plasma edge, the plasma is opaque and is reflecting most of the incident radiation while above it, light is partially transmitted in the form of bulk plasma waves. **c** Experimentally determined plasma frequency $\nu_p$ and scattering rate $\Gamma$ from THz spectroscopy and Drude model as a function of temperature. Center values correspond to the best-fit parameter and error bars are estimated from the fitting errors accounting for experimental uncertainties (see Fig. 1 of supplementary information). The mechanism for tunability of the plasma frequency by thermal activation of the carriers across the band gap of the material is depicted in the inset.

$\nu_p$'s (high temperatures), we recognize the two main resonances associated with the fundamental cavity mode and inter-cavity mode. They both lead to a strong absorption in the reflectivity spectra. As $\nu_p$ decreases, we observe a simultaneous decrease of the two resonance frequencies. This can be understood by noticing that the THz surface plasmons are the collective modes responsible for the emergence of the cavity and inter-cavity resonances. Because surface plasmons only exist within the opacity region of the plasma, the two resonance frequencies remain necessarily upper-bounded by $\nu_p$. This phenomenon enables in turn to make widely tunable THz cavities by simply varying the temperature of the material.

It is possible to explore further the scaling properties of the fundamental cavity resonance by comparing the dimension $s$ of the cavity together with the natural length scale $\frac{c}{\nu_p}$ introduced by the plasma. Indeed, scaling the cavity dimensions at fixed plasma frequency, as done conventionally, is equally achieved here in an elegant fashion by varying the plasma frequency $\nu_p$ while keeping the dimensions of our resonators unchanged. In other words, it is possible to explore the dispersion relation of the cavity resonance by introducing: i) a scaled wave vector $\tilde{k} = \frac{k_x c}{\omega_p} = \frac{c}{2s\nu_p}$ which compares the two characteristic length scales together and ii) a scaled resonance frequency $\tilde{\omega} = \frac{\omega_c}{\omega_p}$ ($\omega_c = 2\pi\nu_c$ and $\omega_p = 2\pi\nu_p$). In Fig. 3d-f, experimental and theoretical dispersions obtained from the parameter free EM-simulation are presented and seen to agree well together. As anticipated, all cavities exhibit a dispersion relation that is characteristic of surface plasmons: a strong departure from the dispersion of the light in the dielectric as well as an asymptotic saturation of the resonance frequencies below the plasma frequency for increasing $\tilde{k}$-vectors. We report in

Fig. 3d-f the effective refractive index of the guided mode $n_{eff} = \frac{k_x c}{\omega_c} = \frac{\tilde{k}}{\tilde{\omega}}$ (grey scale). At the resonance frequency, it quantifies the degree of plasmonic confinement as the ratio between the wavelength $\lambda_0$ of the THz photons in vacuum with the wavelength $\lambda_c$ of the THz photons inside the cavity ($n_{eff} = \frac{\lambda_0}{\lambda_c}$). As the temperature is lowered, we observe that our resonators move deeper into the plasmonic regime of light-matter interaction where $n_{eff}$ rises considerably and where free space THz photons get increasingly confined. At the lowest temperatures, the effective index reaches values in the range $n_{eff} \approx 10-12$ for all three cavities.

## Ultrasmall plasmonic cavity and the nonlocal regime of THz plasmonics

We now seek to determine the highest possible degree of confinement experimentally and realistically reachable with this system. Quite generally, material dissipation is one of the dominant mechanisms that prevents infinite shrinkage of the cavity dimensions. In practice, a subtle balance between unavoidable material dissipation and desired confinement has to be maintained in order to remain as close as possible to a critical coupling condition for efficient coupling of the THz light into the cavity. We successfully managed to achieve this compromise by manufacturing a cavity with dimensions as small as $s = 5.5\,\mu m$, $a = 1.5\,\mu m$ and $d = 0.24\,\mu m$. Close to room temperature, this cavity exhibits sizeable resonant absorption of the incoming THz light of the order of 50% around 1 THz.

The temperature dependence of the reflectivity of the cavity is presented in Fig. 4a. We plot in Fig. 4b its dispersion relation, the dispersion relation computed using a local model for the permittivities

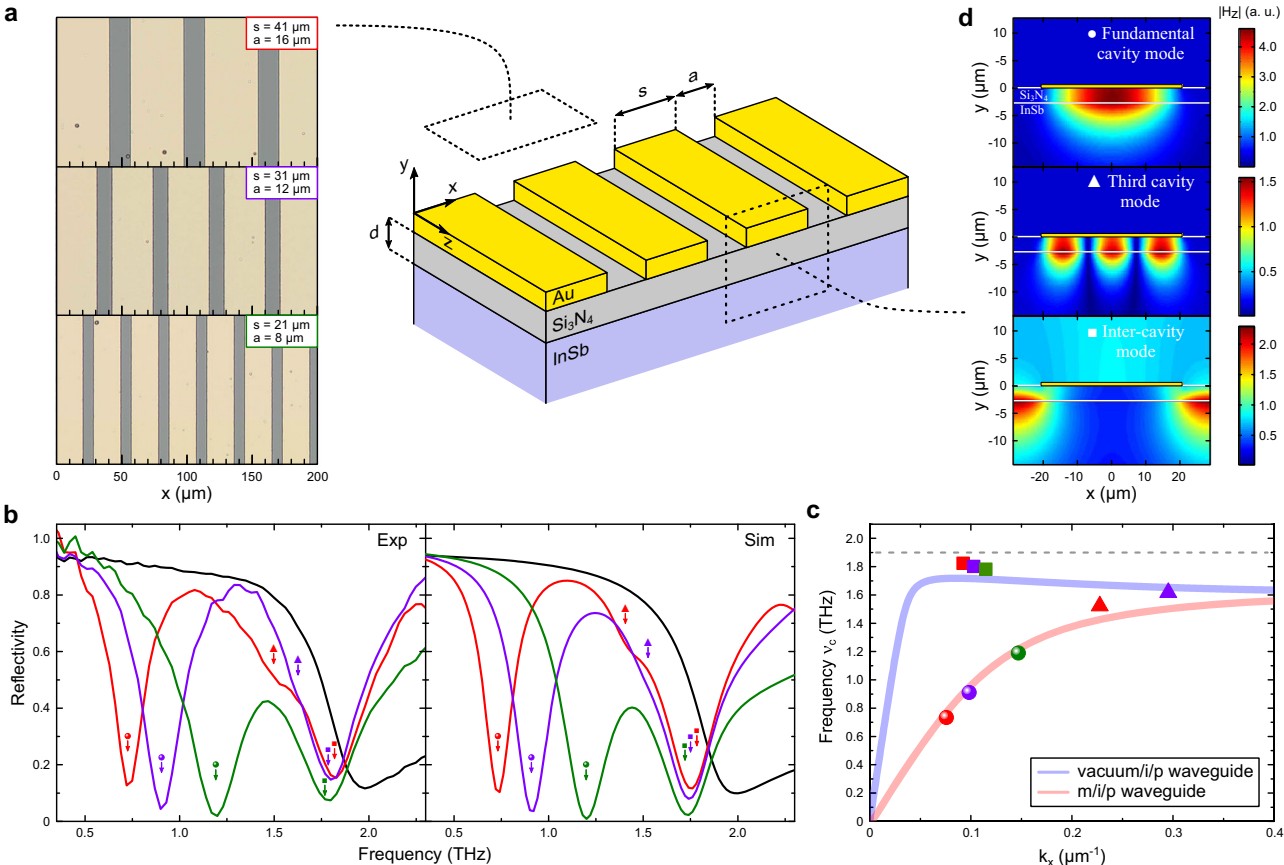

**Fig. 2 | THz plasmonic cavity architecture and experimental results at room temperature. a** Design of the stripe **m/i/p** plasmonic cavities. Sketch of the sample architecture (not on scale) together with the major geometrical parameters $s$ (stripe width), $a$ (inter-cavity spacing) and $d$ (insulator thickness). For all practical purposes, the sample can be considered infinite along $z$ and infinitely periodic along $x$ as the THz spot size is well below the sample size. Also shown are top view optical pictures of the actual samples. **b** Experimental (left panel) and simulated (right panel) reflectivity spectra at T = 290 K for: $s = 41\,\mu m$ sample (red solid line), $s = 31\,\mu m$ sample (purple solid line), $s = 21\,\mu m$ sample (green solid line) and the unstructured bulk InSb (black solid line). Labelling of the different resonances is indicated: fundamental cavity resonances (circles), third cavity resonances (triangles) and inter-cavity resonances (squares). **c** Frequencies $v_c$ of the resonances as a function

of the wavevector $k_x$ of the corresponding modes with the same labelling as in Fig. 1b. Solid lines are the theoretical dispersion relations of: (i) an infinite **m/i/p** waveguide (red solid line) and (ii) an infinite **vacuum/i/p** waveguide (blue solid line). The material parameters used to simulate the dispersion relations are the ones obtained from experimental measurements on the samples at T = 290 K. The grey dashed line corresponds to the plasma frequency $v_p$. **d** Simulated mode profiles of the TM-modes at frequencies corresponding to the three resonant frequencies observed in the reflectivity spectra (simulated for the $s = 41\,\mu m$ sample): the magnitude $|H_z|$ of the magnetic field is obtained for a cut in the $x$-$y$ plane (the structure has translational invariance along $z$). The boundaries between different materials (white solid lines) and the metallic stripe (yellow rectangle) are displayed.

and the refractive effective index $n_{eff}$. We first note that for this ultra-small cavity, plasmonic confinement is enhanced dramatically as $n_{eff}$ reaches values as high as ~ 36. However, when comparing the simulated and experimental dispersion relations, it is clearly seen that there is a deviation between the two which increases as $\bar{k}$ increases. Furthermore, the resonance disappears for wavevectors larger than $\bar{k} \approx 27$. The discrepancy observed in this ultra-high $\bar{k}$ regime suggests the onset of non-local effects[44,46,54–56]. Such nonlocal interactions are brought about by the extreme confinement of the THz cavity field whose typical dimensions become comparable to the characteristic length scales of the electron gas of the THz plasma. Among those, the charge screening length $\lambda_{sc}$ (so-called Thomas-Fermi and Debye screening length in the quantum and classical cases, respectively) plays an important role by introducing an additional convective term for the matter part in Maxwell's equations which is absent within a local EM description. This additional term originates from a hydrodynamic description of the carriers[57]. Physically, longitudinal electric fields can be screened by electrons up to a spatial period $\lambda_{sc}$ given by the electron displacement during a plasma oscillation so that $\lambda_{sc} = v_F/\omega_p$ (where $v_F$ stands for the Fermi velocity in the degenerate regime and for the thermal velocity in the classical limit).

In order to assess quantitatively the validity of this picture, we compare the full frequency resolved absorption spectra of the cavity together with simulations performed in the local and nonlocal regime of light-matter interaction (see methods). For this comparison, we have performed in Fig. 4c a rescaling of the frequencies $v$ by $v_p$ which allows for a convenient examination of the scaling properties of the resonance. In this representation, the local case predicts a resonant absorption which barely shifts as a function of the plasma frequency. On the other hand, by computing the frequency resolved absorption spectra with a single adjustable parameter describing the physics of nonlocality, i.e. the carrier velocity $v_F$, the nonlocal case reproduces very well the strong blue-shift as well as the shape of the resonance that we observe experimentally. Based on this, we plot in Fig. 4d our experimentally determined screening length $\lambda_{exp} = v_F/\omega_p$ and compare it with the Debye screening length $\lambda_{sc} = \sqrt{\frac{3k_B T}{m^*}}/\omega_p$ ($k_B$ being the Boltzmann constant and $m^*$ the effective mass of the carriers) which is expected to describe best the screening of charges in the quasi-classical regime of our dilute electron gas[58]. Here again, a convincing agreement between experiment and theory is observed. Furthermore, the vanishing of the resonance at wavevectors larger than $\bar{k} \approx 27$ can be

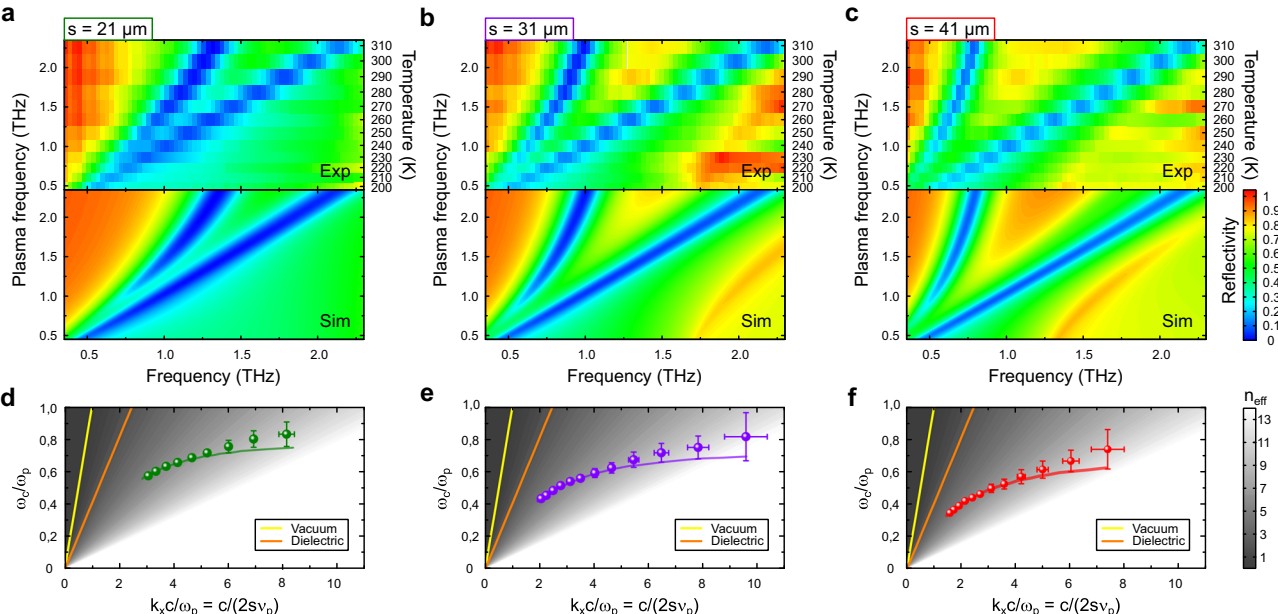

**Fig. 3 | Temperature tuning and scaling properties of the THz plasmonic cavities. a, b, c** Experimental (top panels) and simulated (bottom panels) reflectivity spectra of the three **m/i/p** cavity samples as a function of the bulk plasma frequency (**a**: $s = 21\,\mu m$, **b**: $s = 31\,\mu m$, **c**: $s = 41\,\mu m$). For the experimental measurements, the temperatures corresponding to the plasma frequencies are reported on the right axis of the plots. The color scale indicates the reflectivity. **d, e, f** Experimental (data points) and simulated (solid lines) dispersion relations of the cavities : scaled frequency $\bar{\omega}$ as a function of the scaled wavevector $\tilde{k}$ (see text for details) (**d**: $s = 21\,\mu m$, **e**: $s = 31\,\mu m$, **f**: $s = 41\,\mu m$). The effective refractive index $n_{eff}$ of the guided mode is displayed as a grey scale. The dispersion relations of the light in vacuum (yellow solid line) and in the dielectric (orange solid line) are indicated. $\bar{\omega}$- and $\bar{k}$-error bars are obtained from the propagation of uncertainties on the cavity resonance and plasma frequencies.

well accounted for by Landau damping. The onset of Landau damping takes places when spatial frequencies $q$ of the plasmonic mode satisfy $q > \omega/v_F$. Following the treatment in[59], an estimate of the fraction $F$ of the total power of the mode verifying this condition in our case gives $F = \frac{2}{\pi}n_{eff}\frac{v_F}{c} \approx 15\%$. This leads to an additional scattering channel with scattering rate $\Gamma_{Landau} \approx 0.15$ THz, i.e. of the order of the bulk Drude scattering rate. It is sufficient to suppress completely the resonance and it signals the ultimate limit of plasmonic confinement that is fundamentally reachable.

## Discussion
In view of their future applications, we summarize below the major figures of merit of the THz plasmonic cavities and how they compare with those of metallic-based cavities.

### Ultrasmall mode volumes
We manufactured two dimensional patch cavities having area $A = s \ast s$ (with $s = 5\,\mu m$) and demonstrating similar behavior as the stripe cavities presented above (see supplementary information). For them, we determine the cavity mode volume to be $V_{cav} \approx 4.5\,\mu m^3$ as computed from the Quasi Normal mode approach[60] and $V_{cav} \approx 0.6\,\mu m^3$ by adopting the quantum optics definition used in other works[19]. When normalized to the free space resonating wavelength $\lambda_0$, this leads to volume ratios $V_{cav}/\lambda_0^3$ reaching values down to $1.0\ast10^{-7}$ and $1.3\ast10^{-8}$ respectively. These values are found to be competitive with state-of-the-art ultrasmall metallic cavities based on more complex architectures[19,22].

### Record breaking small cavity footprints
The cavity footprint, defined by its area $A = s \ast s$, leads to normalized surface ratios $A/\lambda_0^2$ down to $2\ast10^{-4}$. This outperforms by more than two orders of magnitude the footprints of conventional **m/i/m** and split-

ring type THz cavities and by an order of magnitude the smallest 3D-lumped THz resonators[21,22].

### Limited $Q$−factors
Typical $Q$−factors are of the order of 3 to 4 and found to be dominated by the non-radiative plasmonic losses. These $Q$−factors are equivalent to other ultrasmall metallic cavities and, more generally, comparable with most metallic resonators having $Q$−factors in the range of 10 typically.

### Ultrastrong light-matter coupling in the few-electrons limit
Considering the replacement of the insulator by a quantum well heterostructure consisting of $N$ electrons having an intersubband transition in resonance with the cavity mode, the vacuum Rabi splitting $2\Omega$ can be expressed in the present context as $2\Omega = \sqrt{f_{12}}\sqrt{\frac{Ne^2}{m^\ast\epsilon\epsilon_0 ds^2}}$, where $f_{12} \approx 1$ is the oscillator strength of the lowest $1 \rightarrow 2$ transition for deep rectangular quantum wells and $m^\ast$ the electron effective mass[61]. Taking $2\Omega \gtrsim \Gamma$ as the criterion for observation of the Rabi splitting ($\Gamma$ being the width of the cavity mode), we estimate that the ultrastrong coupling regime could be reached with GaAs-based quantum wells at $N \sim 2000$ electrons. For InSb-based quantum wells, this value is lowered to $N \sim 600$, making the plasmonic cavities promising candidates for THz cavity-QED experiments in the few-electrons limit.

### Excellent coupling efficiencies
As in **m/i/m** cavity arrays[16], geometrical tuning of the radiative and non-radiative $Q$−factors has allowed to obtain a very good contrast for the resonances in our experiment. This demonstrates an excellent coupling efficiency to external THz radiation suitable for THz detection applications for instance.

In conclusion, using narrow-gap semiconductor InSb, we demonstrated the realization of ultrasmall and widely frequency-tunable THz

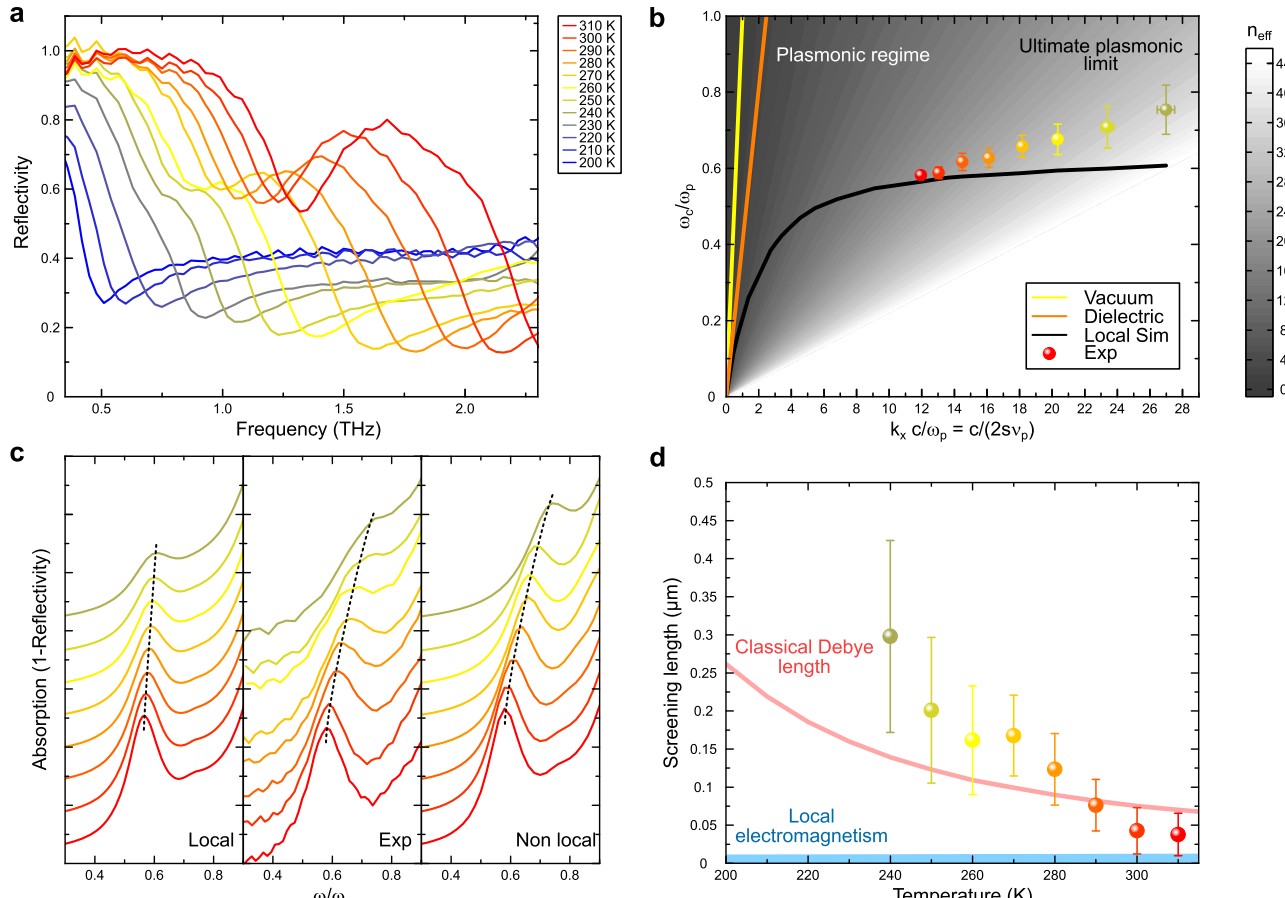

**Fig. 4 | Ultrasmall plasmonic cavity and the nonlocal regime of THz plasmonics.** Results for the smallest stripe **m/i/p** cavity sample ($s = 5.5\,\mu m$). **a** Reflectivity spectra as a function of frequency between 200 K and 310 K. **b** Plasmonic dispersion of the cavity: $\tilde{\omega} = \omega_c/\omega_p$ as a function of $\tilde{k} = k_x c/\omega_p$. Experiment (colored data points) and simulation of the dispersion from the local model (black solid line) are compared together. Light dispersion in vacuum (yellow solid line) and in the dielectric (orange solid line) is reported. Color code of the data is used to label the temperature with the same convention as in Fig. 4a. $\tilde{\omega}$- and $\tilde{k}$-error bars are obtained from the propagation of uncertainties on the cavity resonance and plasma frequencies.
**c** Comparison of the frequency resolved absorption spectra (=1-Reflectivity): simulations (local model: left panel/ nonlocal model: right panel) and experiment (middle panel). Spectra at different temperatures are offset for clarity. The

frequency axis is rescaled by the plasma frequency to compare the scaling of the resonance in each case. Dashed lines are a guide to the eyes indicating the position of the resonance. Color code of the data is used to label the temperature with the same convention as in Fig. 4a. **d** Temperature dependence of the screening length determined by fitting the non-local EM-simulations to the experiment (see text). Also shown is a comparison with the Debye screening length (pink solid line) and the infinitely small screening length describing the case of local EM-interactions (blue solid line). Color code of the data is used to label the temperature with the same convention as in Fig. 4a. Center values correspond to the best-fit parameter and error bars are estimated from the fitting errors accounting for experimental uncertainties (see Fig. 4 of supplementary information).

cavities based on a surface plasmonic mechanism. We discovered that the plasmonic confinement can be taken to its most ultimate limit related to EM-nonlocality and Landau damping at frequencies as low as 1 THz. In this regime, plasmonic-based cavities exhibit figures of merit that rival or even surpass metallic-based ones, allowing them to be readily operational for similar applications. Beyond this, the most remarkable asset of THz cavities taking advantage of a plasmonic mechanism resides in the large tunability of the plasmonic medium with both static and dynamic external parameters, and hence in the tuning of the THz cavity photon properties. On this basis, we foresee two major applications that extend the current scope of THz cavities and would be challenging to achieve with standard approaches. The first one is the ability to create multi-purpose resonant THz metasurfaces and obtain, *from a single device*, various functionalities that can be toggled on-demand as opposed to designed on a case-by-case basis. The second and most far-reaching application corresponds to the realization of novel types of ultrastrong light-matter coupling experiments. Thanks to the plasmonic tunability, our approach sets the stage for the exploration of the ultrastrong coupling regime of THz cavity

electrodynamics where light-matter interactions can be tailored. This includes, but is not restricted to, chiral and non-reciprocal light-matter couplings for instance, and represents an uncharted path which will open up many bright perspectives in this field.

## Methods
### Sample manufacturing
Plasmonic cavity samples were manufactured starting from a $500\,\mu m$ thick, < 100 > oriented, bulk wafer of InSb commercially available (MTI Corporation). The InSb wafer was nominally N-type undoped. We determined a residual extrinsic doping concentration to be ~ $10^{14}\,cm^{-3}$ from the measurement of the plasma frequency $\nu_p \approx 0.2\,THz$ which remained independent of the temperature between 4 K and 150 K. The thermally activated (intrinsic) regime of the semiconductor manifested for temperatures above 150 K. Each sample consisted of a ~ 5 $mm*5\,mm*0.5\,mm$ InSb substrate dice-cut from the wafer. An insulating layer of $Si_3N_4$ was then deposited via plasma enhanced chemical vapor deposition. The top metallic parts of the stripe and patch cavities were realized via photolithography followed by metal deposition of Ti(15 nm)/Au(200 nm). The geometrical and optical parameters of

the constitutive elements of the samples have been experimentally determined and are reported in the supplementary information.

## THz spectroscopy

Spectroscopy of the cavities was performed with a time-domain THz spectrometer driven by an ultrafast Ti:Sapphire oscillator. THz generation and detection were achieved via a photoconductive emitter (Tera-SED) and a 1 mm thick < 110 > ZnTe crystal, respectively, allowing for spectroscopic coverage from 0.2 THz to 2.5 THz. Delayed THz pulses originating from the optical components of the setup were eliminated by windowing the main THz pulse reflected off the samples. The time domain signals were Fourier transformed and analyzed in both their squared amplitudes (power reflection) and phases. All reflectivity measurements were performed at normal incidence on samples located inside a temperature controlled optical cryostat.

To obtain an absolute measurement of the reflectivity, we used the THz pulse reflected off a metallic surface as a reference. For the striped cavity samples, the spectra were obtained by dividing the power reflectivities measured with the THz electric field polarized perpendicular to the metallic stripes (TM-polarization, optical activity of the cavity modes) and parallel to the stripes (TE polarization, inactivity of the cavity modes). The TE-polarized reflectivity is flat and featureless with reflectivity over 98% in the frequency range investigated, allowing to use it as a reference for the measurement. For the bulk InSb where this method is not applicable, larger samples were half-coated with a Gold reference. In all measurements, the THz spot size ( ~ 2 mm in diameter) is well below the size of the manufactured samples, so that the samples can be considered infinite in the plane containing the interface (x-z plane in Fig. 2a). The overall uncertainty on the amplitude of the electric field of the reference THz pulse in the reflection geometry is estimated to be of order of 5% by comparing different runs of measurements together.

## Numerical simulations

Numerical simulations of the cavities were performed with the Rigorous Coupled Wave Analysis method[62,63]. The reflectivity of stripe cavities was determined with the same procedure as done in the experiments by dividing the power reflectivities obtained for TM-polarization (electric field polarized perpendicular to the stripes) and TE-polarization (electric field polarized parallel to the stripes). The parameters used for the simulations are determined from experimental measurements performed on the samples (see supplementary information). The permittivity of Gold at THz frequencies was taken from[50]. As a result, the simulations performed in the local regime of light-matter interaction are parameter-free. EM-modelling in the nonlocal regime of light-matter interaction was implemented following the procedure described in[64] and taking into account background polarizability[54] (see also supplementary information). Computations were performed with a single input parameter describing the physics of nonlocality: the carrier velocity $v_F$. Regression over an error function was then applied in order to determine $v_F$ that correspond to the best fits between experimental and simulated spectra over the frequency resolved absorption spectra in the range $\nu = [0.3\nu_p, 0.9\nu_p]$. Uncertainties in $v_F$ were determined from the comparison of the position and the shape of the resonance taking into account the uncertainties of the experimental spectra and material parameters.

## Data availability

The data presented in this study are available from the corresponding authors upon request.

## Code availability

The numerical simulations are done with the RETICOLO software that can be found online at https://zenodo.org/record/4419063.

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

## Acknowledgements

Y.L. acknowledges support from Agence Nationale de la Recherche (grant n° ANR- 17-CE30-0011-11) and Labex PALM (grant n° ANR-10-LABX-0039-PALM). Y.L. would like to thank D. Hagenmuller for fruitful discussions.

## Author contributions

Y.L. conceived the experiment and supervised the project; I.A. manufactured the samples with contributions from R.G., D.D. and J.B.; I.A. performed the experiments, collected and analyzed the data; additional characterization of material properties were made by I.A. with contributions from T.G.; The results were analyzed and discussed by all co-authors; J.P.H. and J.J.G provided theoretical support and performed the numerical simulations; Y.L. wrote the manuscript with contributions from I.A., S.H., L.P., J.J.G. as well as other authors.

## Competing interests

The authors declare no competing interests.

## Additional information

**Peer review information** : *Nature Communications* thanks the anonymous reviewers for their contribution to the peer review of this work. A peer review file is available.

