## [Peer Review File · Nature Communications]

Ultrasmall and tunable TeraHertz surface plasmon cavities at the ultimate plasmonic limitEditorial Note: This manuscript has been previously reviewed at another journal that is not operating a transparent peer review scheme. This document only contains reviewer comments and rebuttal letters for versions considered at *Nature Communications*. Mentions of prior referee reports have been redacted.

REVIEWERS' COMMENTS

Reviewer #1 (Remarks to the Author):

[redacted]. I recommend publication.

My remaining comment is with the conclusion and some of the discussion within. The justification for strong coupling is made in terms of the Purcell factor. While the two are related, it is slightly confusing since the Purcell factor picture is valid in the weak coupling regime. I'm not quite sure where actual Purcell effect would be relevant – making a THz quantum-cascade LED? The application of (ultra) strong-coupling is more plausible then – perhaps estimation of a Rabi-splitting in an ISB system would be a more relevant metric.

Reviewer #3 (Remarks to the Author):

[redacted].

Response to referees

We would like to thank the two referees for their very positive feedback and for recommending the publication of our manuscript in Nature Communications.

Below, we answer to the referees' comments.

REVIEWERS' COMMENTS

Reviewer #1 (Remarks to the Author):

[redacted]. I recommend publication.

My remaining comment is with the conclusion and some of the discussion within. The justification for strong coupling is made in terms of the Purcell factor. While the two are related, it is slightly confusing since the Purcell factor picture is valid in the weak coupling regime. I'm not quite sure where actual Purcell effect would be relevant – making a THz quantum-cascade LED? The application of (ultra) strong-coupling is more plausible then – perhaps estimation of a Rabi-splitting in an ISB system would be a more relevant metric.

Our answer: [redacted]. We also acknowledge the suggestion made so as to improve the discussion on the ultrastrong coupling regime for these cavities.

Accordingly, we removed the figure of merit related to the Purcell factor and estimated instead the Rabi-splitting of an ISB transition in quantum well heterostructures that would replace the dielectric insulator. In particular, taking the point of view of ultrastrong coupling regime in the few-electron limit, we sought to determine the number of ISB electrons necessary to obtain an observable Rabi splitting in our system. Following the approach in [Y. Todorov et al. PRL 105 196402 (2010)], we estimate that the Rabi-splitting should be observable at around 2000 electrons with GaAs quantum wells and 600 electrons with InSb quantum wells. We added this discussion and this figure of merit in the revised version of the manuscript.

Reviewer #3 (Remarks to the Author):

[redacted].

Our answer: [redacted].

The changes made from the previous version of the manuscript are highlighted in red color in this revised version.

Changes made to the manuscript according to the referee's comments:

- 1) We removed the discussion on the Purcell factor and discussed instead the ultrastrong coupling regime of an ISB transition.

Other changes have been made to comply with Nature editorial policy: a shortened abstract, figure titles, error bars description, etc.